# In Vitro Culture of Chicken Circulating and Gonadal Primordial Germ Cells on a Somatic Feeder Layer of Avian Origin

**DOI:** 10.3390/ani10101769

**Published:** 2020-09-30

**Authors:** Agata Szczerba, Takashi Kuwana, Michelle Paradowska, Marek Bednarczyk

**Affiliations:** Department of Animal Biotechnology and Genetics, Faculty of Animal Breeding and Biology, UTP University of Science and Technology, Mazowiecka 28, 85-084 Bydgoszcz, Poland; kuwanat@gmail.com (T.K.); michelle.paradowska@utp.edu.pl (M.P.); marbed13@op.pl (M.B.)

**Keywords:** primordial germ cells, gPGCs, cPGCs, chicken, embryo, feeder cell layer

## Abstract

**Simple Summary:**

Chicken primordial germ cells are specialized cells that are formed outside the developing embryo, from where they migrate into the gonad, and give rise to the gametes. There are two types of those cells: Circulating blood primordial germ cells and gonadal primordial germ cells. They can be isolated only in low numbers from the bloodstream or gonads of donor embryos. Hence, efficient in vitro cultivation systems are required to increase their quantity through proliferation. Here, we provide a single culture system that can be used to cultivate both cell types. We also present a novel, easy-to-train, and non-invasive method to identify live primordial germ cells in the culture. In this analysis, chicken primordial germ cells obtained from embryonic blood or gonadal regions were cultured in vitro on feeder cell layers derived from embryonic chick cells. The use of the chicken origin feeder layer allowed reducing the xenogenic animal factors in the culture. We demonstrated a feasible and cost-efficient technique to routinely assess the cultivated primordial germ cells on the basis of their morphological characteristics and using the optical features of cells in darkfield illumination. This method is especially useful to distinguish primordial germ cells during co-cultivation with other cell types.

**Abstract:**

The present study had two aims: (1) To develop a culture system that imitates a normal physiological environment of primordial germ cells (PGCs). There are two types of PGCs in chicken: Circulating blood (cPGCs) and gonadal (gPGCs). The culture condition must support the proliferation of both cPGCs and gPGCs, without affecting their migratory properties and must be deprived of xenobiotic factors, and (2) to propose an easy-to-train, nonlabeling optical technique for the routine identification of live PGCs. To address the first aim, early chicken embryo’s feeder cells were examined instead of using feeder cells from mammalian species. The KAv-1 medium at pH 8.0 with the addition of bFGF (basic fibroblast growth factor) was used instead of a conventional culture medium (pH approximately 7.2). Both cPGCs and gPGCs proliferated in vitro and retained their migratory ability after 2 weeks of culture. The cultivated cPGCs and gPGCs colonized the right and/or left gonads of the recipient male and female embryos. To address the second aim, we demonstrated a simple and rapid method to identify live PGCs as bright cells under darkfield illumination. The PGCs rich in lipid droplets in their cytoplasm highly contrasted with the co-cultured feeder layer and other cell populations in the culture.

## 1. Introduction

An efficient in vitro system for the cultivation of avian primordial germ cells (PGCs) is required for routine implementation of PGC-based programs to conserve endangered species [1,2,3,4] as well as for genome editing [5] and transgenic animal technologies [6].

PGCs are the only cells in developing embryos that have the potential to transmit genetic information to the next generations. This potential exists even when PGCs are isolated from donors, manipulated in vitro, and reintroduced into recipient embryos [7]. However, several factors limit the use of avian PGCs on a wide scale. One of the most problematic technical issues is the limited number of PGCs that can be obtained from a single embryo [8,9]. Although PGCs can be sourced either from blood as circulating PGCs (cPGCs) or from gonads (gPGCs) at certain embryonic stages, the recoverable cell population number is low and different, for both types of PGCs contain lipid droplets and accumulated glycogen. Both cPGCs and gPGCs stain with anti-SSEA-1, anti-EMA-1, anti-CVH, anti-integrin β1, and anti-CEACAM antibodies [10,11].

To enable the biotechnological use of a limited number of isolated PGCs, it is important to increase their populations by in vitro propagation. Park and Han [12] were the first who documented the effective long-term culture of gonadal PGCs. The cells were isolated on day 5.5 of embryo incubation (at the 28 stage Hamburger and Hamilton (H&H) of development) [13] and cultured with SCF (stem cell factor), LIF (leukemia inhibitory factor), bFGF (basic fibroblast growth factor), IL-11 (interleukin 11), and IGF-1 (insulin-like growth factor 1) on the chick embryonic fibroblast layer at pH 7.2. Naito [14] also used a feeder cell layer to propagate the growth of cPGCs obtained from chick circulating blood. The long-term culture of cPGCs was also performed by Van De Lavoir et al. [15], with the addition of FGF and SCF to a conditioned medium. These cPGCs were cultivated on STO (mouse, Sandos inbred mouse (SIM)-derived 6-thioguanine- and ouabain-resistant) or BRL (buffalo rat liver) feeder cells. Nevertheless, the propagation of female PGCs in their system was less effective than that of male PGCs [15,16]. Similarly, in the method of Tonus et al. [17], all the resulting PGC lines tended to be devoid of female PGCs.

cPGCs can be collected from the embryo bloodstream during H&H stages 13 to 16 and placed into culture plates as “whole blood” isolates. One of the arguments for the use of gPGCs is their relatively higher number that can be extracted from gonadal ridges [18]. Moreover, less operatory skills are needed to isolate gonads than to manipulate blood arteries in the embryos. Yet, it cannot be definitely determined which PGC type should be preferred over another to successfully achieve various technological goals. Kim and Han [19] emphasize a need to gain a deeper knowledge of epigenetic mechanisms, which might be affected not only by the culture conditions but also by somatic factors of recipient surrogates. Marker-assisted sorting techniques have been developed to fractionate PGCs from the isolated cell cohorts and can be used for both gPGCs and cPGCs [20].

Here, we provide a single culture system that can be used to cultivate both cell types—gPGCs and cPGCs. We assumed that the culture condition based on a feeder layer of avian origin and medium optimized to pH 8.0 would imitate the natural environment of PGCs and thus support their proliferation and migration. We also present a novel, easy-to-train, and noninvasive method to identify live PGCs in the culture.

## 2. Materials and Methods

### 2.1. Fertilized Eggs and Incubation

In every planned experiment, 30 fertilized eggs from the Ross 308 crossbred chicken were purchased from Drobex-Agro Sp. Zoo. (Solec Kujawski, Poland). Eggs were incubated at 38 °C and 65% relative humidity for 72 h to obtain gPGCs and fibroblasts for the feeder layer. Both types of cells were obtained from embryos at 18 stage of H&H. Fertilized eggs were also incubated at 38 °C and 65% relative humidity for 53 h to reach H&H stages 14–15 to obtain cPGCs.

All experimental procedures were conducted according to the guidelines for the care and use of experimental animals of University of Science and Technology. The experimental protocols were approved by the Local Ethical Committee for Animal Experiments in Bydgoszcz.

### 2.2. Preparation of Feeder Cells

A feeder layer for cultivating chicken PGCs was used in the present analysis. Cells from the chick embryos were collected to obtain the feeder layer. The posterior region of the vitellin artery of the embryos was cut with micro-scissors. The artery was then washed twice with phosphate-buffered saline (PBS, SH30256.02, HyClone^TM^, Logan, UT, USA). The excised tissues were rinsed with Ca^2+^- and Mg^2+^-free PBS (PBS(-); 52321C, Sigma-Aldrich, St. Louis, MO, USA) and submerged into 150 μL of 0.1% Trypsin-0.02% EDTA in PBS(-) in a plastic Petri dish (93060, TPP, Switzerland) at room temperature. After 20 min, 150 μL of KAv-1 medium [21] was added to disperse the tissue and obtain a cell suspension. Next, it was transferred to a 1.5 mL Eppendorf tube and filled to a volume of 1 mL with KAv-1 medium. The suspension was centrifuged at 400× *g* for 5 min, and the supernatant was removed. The remaining pellet was suspended in 2 mL KAv-1 medium and transferred into a well of a 48-well plate (Costar 3526, Corning Inc., New York, NY, USA). The well was coated earlier with collagen from rat tail and secured with a sealing film (Platemax^®^ AxySeal Sealing Film, PCR-SP, Axygen Inc., Union City, CA, USA). The cells were incubated at 37.8 °C. The medium change was performed every 2 to 3 days.

In the next step, cultured feeder cells were treated with 10 μg/mL mitomycin C (MMC) in KAv-1 medium at 38 °C and washed two times with fresh KAv-1 medium after 2 to 3 h. The cells were also treated with 0.1% Trypsin-0.02% EDTA in PBS(-) for 8 min at 38 °C. Next, the cells were suspended in the solution with the same volume of fresh KAv-1 medium. The obtained cell suspension was centrifuged at 800× *g* for 3 min at room temperature. The supernatant was removed, and the pellet was suspended in the fresh KAv-1 medium. The cells were cultured in a plastic flask (12.5 cm^2^, 353018, Falcon, Corning, New York, NY, USA) with 10 μM/mL Y-27632 (SCM075, Sigma-Aldrich, St. Louis, MO, USA) until use.

### 2.3. Chick Embryo Extracts

Fertilized eggs from chicken breed Ross 308 were incubated for 5 days. The chick embryos were isolated from the yolk and washed with physiological saline. Each embryo was homogenized with the 800 µL KAv-1 medium. The received suspension was centrifuged for 30 min at room temperature. The supernatant was collected for further use. The obtained chick embryo extract (CEE) was filtered with a 0.45-µm pore size membrane filter (99745, TPP, Trasadingen, Switzerland) and stocked at −20 °C until use.

### 2.4. Preparation of Rat Tail Collagen

A rat tail was used to obtain collagen. The adult rat tail was washed two times with PBS, and the skin from the tail was removed. The tail was then cut into 1-cm sections with clean scissors. By using forceps, the tendon fibers were detached and collected in fresh PBS. The collected fibers were washed with pure water (03-055-1A, BI, Beit-Haemek, Israel) and sterilized by soaking in 70% ethanol for 20 min. They were also submerged in 99.6% ethanol for 10 min and then washed twice with sterile pure water. Finally, the tendon fibers were immersed in a solution of 0.1% acetic acid in pure water for 1 week at 4 °C. Following this, the supernatant was used to coat the bottom of the wells of multi-well plates.

### 2.5. Culture of PGCs

#### 2.5.1. Culture Medium

In the present study, a basic KAv-1 medium containing 5% fetal bovine serum (F2442-100ML, Sigma-Aldrich) and 5% chicken serum (C5405-500ML, Sigma-Aldrich) was used as a standard medium. This medium was adjusted to pH 8.0 [21]. The modified KAv-1 medium (mKAv-1) was also used, which contained 5% Knock Out Serum Replacement (KO-SR; 10828028, Gibco by Life Technologies, Waltham, MA, USA), 10 ng/mL b-FGF (F0291-25UG, Sigma-Aldrich), and 20 µL/mL of the earlier obtained CEE.

#### 2.5.2. Culture of gPGCs

The gPGCs were obtained from embryos at stages 17 to 18 H&H. The embryos (*n* = 15 per experiment) were washed with physiological saline. Next, the gonadal regions were excised using micro-scissors. The gonadal region at H&H stages 17 to 18 comprises the posterior embryonic body region of vitelline arteries. The tissues were collected in an Eppendorf tube and washed with physiological saline to remove yolk granules. They were then treated with 500 µL of PBS(-) for 3 min at room temperature and centrifuged at 800× *g* for 2 min. The supernatant was then removed. The remaining tissues were soaked in 500 µL of 0.1% Trypsin−0.02% EDTA in PBS(-) and incubated at 38 °C. After 8 min, 500 µL of KAv-1 medium was added to the tube, and the tissues were gently dispersed. The suspension was then centrifuged at 800× *g* for 2 min. The supernatant was removed, and the remaining cells were dispersed. Finally, the cells were seeded into a 12.5 cm^2^ culture flask (Falcon, Corning, New York, NY, USA) with 3 mL of modified KAv-1 medium. The cells obtained from the gonadal regions were cultured at 38 °C. Half of the medium was replaced with a fresh medium every 2 days.

The gPGCs were subcultured when they completely covered the surface of the culture flask, usually after 3 days of culture and every 3 days thereafter. The proliferated gPGCs were collected using 0.02% EDTA in PBS(-), which was gently pipetted on the cells. To collect gPGCs that were loosely attached to the somatic cells, the culture flask was rinsed twice with 2 mL of PBS(-). The obtained solution of gPGCs was stocked in a 15-mL plastic centrifuge tube (Corning, USA). The remaining gPGCs in the flask were additionally treated with 2 mL of 0.02% EDTA in PBS(-) for 10 min at room temperature. The solution with remaining clusters of gPGCs was collected into the same 15 mL plastic centrifuge tube as described previously. The next step was centrifugation at 800× *g* for 3 min. The cell pellet was suspended in 1 mL of 0.1% Trypsin−0.02% EDTA PBS(-) and kept at 38 °C. After 6 min, 1 mL of fresh KAv-1 medium was added and suspended to obtain a single cell suspension. The cells were centrifuged once again in 800× *g* for 3 min at room temperature and then seeded on MMC (MMC; M4287-2MG, Sigma-Aldrich)-treated feeder cells.

#### 2.5.3. Culture of cPGCs

To obtain cPGCs, the first step was to extract embryonic blood from 10 embryos at embryonic stages 14 to 15 H&H. The blood was withdrawn using a fine glass micropipette and pooled in a 1.5-mL Eppendorf tube with 500 mL KAv-1 medium. The blood was then centrifuged at 800× *g* for 3 min at room temperature. The collected cells were labeled with the fluorescent dye PKH26-GL (Z-PKH26-GL, Zynaxis, Malvern, PA, USA) according to the manufacturer’s instructions. Next, the cells were washed twice with fresh KAv-1 medium and seeded on a chicken feeder cell layer. When the cPGCs started to proliferate in a primary culture, they were passaged (as described for gPGCs in Section 2.5.2) and continuously co-cultured on the feeder cells.

### 2.6. Migratory Abilities of Cultured PGCs to the Recipient Gonads

To determine the migratory ability of pre-cultivated gPGCs and cPGCs, the cells were detached from the feeder layer by using PBS(-) to rinse the cells, followed by treatment with 0.02% EDTA in PBS(-) for 5 min at room temperature. The dispersed cells were labeled with the fluorescent dye PKH26-GL according to the manufacturer’s instructions. Next, 500–800 labeled cells were injected into each embryo’s bloodstream at H&H stages 14 to 16. The cells were injected through a 5 to 10 mm window in the eggshell of the recipients. The windows in the eggshell were sealed with an adhesion tape (#800, Scotch, St. Paul, MN, USA) and additionally sealed on the edges with glue (E301, Elmer’s School Glue, Westerville, OH, USA) to avoid detachment of the tape. The recipient eggs were further incubated at 38 °C for 5 to 6 days. To check the presence of PGCs, the recipient embryos were removed from the eggs and washed with PBS. The gonads were removed from the embryos in PBS. The presence of the labeled cells in the gonads was verified using a fluorescent microscope (DMI8, Leica, Wetzlar, Germany).

## 3. Results

### 3.1. Identification of Fresh and Cultured PGCs

In this study, we present a new criterion to identify PGCs. We used darkfield illumination as shown in Figure 1. PGCs can be easily identified from lipid compounds and deposits in their cytoplasm. PGCs are distinguished as bright cells contrasting with the surrounding cells.

### 3.2. Culture of PGCs

#### 3.2.1. Culture of PGCs Collected at Stage 18 H&H

The gPGCs were subcultured every 4 to 5 days. The morphological features after 1 and 2 weeks of culture showed the same characteristics as those of the freshly isolated PGCs (Figure 2a,b; upper images). The cells proliferated and formed small clusters on the somatic feeder cell layer. In the present study, their doubling time was approximately 84 h gPGCs that could be easily identified as bright cells in darkfield illumination at all times (Figure 2a,b; lower images).

#### 3.2.2. Culture of cPGCs at Stages 14 to 15 H&H

The medium change must be performed on the following day after seeding cPGCs on the feeder cell layer to reduce the number of embryonic blood cells that did not adhere to the feeder layer. Consecutive medium changes were performed every 2 days. After the 4th medium change (7th day of incubation), clusters of proliferated cPGCs were observed (Figure 3). Only a few remaining embryonic blood cells were found. Small clusters of cPGCs were observed by PKH26 staining. These cells had the same morphological features as the bright cells observed under darkfield illumination (Figure 3).

### 3.3. Migratory Ability of Cultured PGCs to the Recipient Gonads

Both types of cells—gPGCs and cPGCs were cultured and then injected into the bloodstream of the recipient embryos at H&H stages 13 to 16. This was performed to confirm whether the pre-cultivated PGCs retained their ability to migrate into the recipient embryonic gonads.

#### 3.3.1. Cultured gPGCs

After 15 days of culture, the gPGCs were diluted with KAv-1 medium to obtain 800 g PGCs/µL, and 3 µL of this solution was injected into the recipient’s peripheral vein at stages 13 to 16 H&H. The embryonated eggs with the injected PGCs were further incubated in the egg incubator at 38 °C. Ten out of 14 recipient (71%) embryos developed normally after 5 or 6 days as shown in Table 1 and Table 2. As shown in Table 1, the injected PGCs that were cultured were incorporated into gonads of all survived embryos (after 5 days, *n* = 4; after 6 days, *n* = 6). Most of the injected gPGCs were found in the left gonad and not in the right gonad. Szczerba et al. [22] observed that after the formation of the posterior vitelline vein, the area of blood vessels in the left gonad increases compared to that in the right gonad. This is due to the formation of the posterior vitelline vein only on the left side of the embryo. The bloodstream in the capillary network on the left side can, therefore, flow directly into the vein, while the bloodstream on the right side of the network cannot flow into any other vein. The older the embryo is, the more is the development of the circulatory system, which may be the cause of higher mortality.

The collected gonads were dispersed by the trypsin treatment to confirm the cytological features of the incorporated labeled cells. As shown in Figure 4, the dispersed PKH26-labeled cells had retained the morphological feature of gPGCs.

The gPGCs were seeded and cultured for at least 7 days. One day after seeding the cells, they were counted manually from many fields of view every day for 7 days by using the Leica DMI8 microscope and darkfield illumination. PGCs were visible as bright cells distinctive from other cell types. This method enabled us to count the cells rapidly and easily without using proliferation assays. Figure 5 shows that the cells proliferated and the fold change increased every day. The gPGCs did not proliferate rapidly because we obtained only three cells from one cell in 7 days.

#### 3.3.2. Cultured cPGCs

After 15 days of culture, the cPGCs were diluted with the KAv-1 medium to obtain 800 cPGCs/µL, and 3 µL of this solution was injected into the recipient’s peripheral vein at stages 13 to 16 H&H. The embryos with the injected cPGCs were further incubated in the egg incubator at 38 °C. Eleven out of 20 (55%) treated embryos developed normally after 5 days as shown in Table 2. Nine out of these 11 (82%) embryos with injected cPGCs that were incorporated into the gonads survived.

All the embryonic gonadal tissues were excised with micro-scissors and assessed for the presence of the injected cultured gPGCs in their tissues. The collected gonads were also dispersed using the trypsin treatment to confirm the cytological features of the incorporated labeled cells. The left column of Figure 6 shows the right male gonad under the microscope light and the same gonad using a fluorescent microscope and the dye PKH26.

The cPGCs were seeded and cultured for at least 7 days. One day after seeding the cells, they were counted manually from many fields of view every day for 7 days using the Leica DMI8 microscope and darkfield illumination. cPGCs were visible as bright cells distinctive from cells of the other types. Figure 7 shows that the cells proliferated and their fold change increased every day. Compared to PGCs obtained from chick embryo gonads, the fold change of cPGCs was higher. Within 7 days, the cells proliferated from one cell to 28 cells.

## 4. Discussion

Here, a noninvasive and simple method was proposed to distinguish PGCs from other cells in the culture (P.434242 patent application number). The PGCs are characterized by the presence of many lipid droplets in their cytoplasm, a relatively large-sized nucleus, and a spherical shape (Figure 1). Even when using a conventional bright-field light microscope, it may be problematic to identify PGCs in heterogeneous populations, and these cells may be confused with other structures (even with lipid droplets sometimes). The cell structural features generate a specific optical condition that enable their illumination-based identification. Therefore, the rapid method proposed here may increase the efficiency of culture evaluation of PGCs.

In this analysis, chicken PGCs obtained at stages 14 to 15 and 18 H&H from embryonic blood or gonadal regions were cultured in vitro on feeder cell layers derived from embryonic chick cells. The use of the chicken origin feeder layer allowed reducing the xenogenic animal factors in the culture. Here, in contrary to other studies, the chick PGC culture did not require more commonly used xenogenic feeder cell layers such as mouse (STO), rat (buffalo rat liver [BRL]), or conditioned medium with rat fibroblasts [15,17,23,24]. Here, the conditions were established for maintaining PGCs using the KAv-1 medium at pH 8.0 instead of KO-DMEM at pH 7.2. Chicken serum (CS), KOSR, and bFGF were the other substances used. According to Van De Lavoir et al. [15], CS, SCF, human FGF, and BRL, which produce LIF, may be very important factors in the culture of PGCs. They are known as factors that cause proliferation of the chick PGCs. Woodcock et al. [2] also used a medium containing CS or serum-free medium containing ova transferrin for their research. This allowed minimizing the use of xenobiotics in the culture. Despite the high pH, there were no considerable issues in the growth of PGCs. According to Kuwana et al. [21], the embryonic blood pH in situ between 2 and 11 days of incubation is approximately 8.0. This implies that in such pH, the functioning of chicken PGC is normal; they proliferate and migrate. The outcome of this study indicates that culture conditions such as pH around 8.0 and the presence of an avian fibroblast layer are similar to the environment that occurs naturally in developing chicks. These conditions also have been indicated in other studies [12,25,26,27].

Several difficulties are encountered while culturing PGCs. The first one is the issue of a very limited number of PGC in situ, which directly poses technical issues to isolate a sufficient founder population. It appears that the most practical method of obtaining PGC is to isolate them from developing gonads of 5- to 7-day-old embryos [18]. This approach allows us to obtain a relatively high amount of viable PGCs in a short time. The disadvantage of this method is the risk of contamination with other types of cells. This shows that no single method is good enough to obtain and cultivate both types of cells. Here, we collected gPGC at H&H stages 17 to 18 from somatic cells of future gonadal regions. The collected gPGCs proliferated and formed clusters on the somatic fibroblast layer. Their morphological features were the same as those of cPGCs in the bloodstream (Figure 1 and Figure 2). The cPGCs had the same morphological characteristics after 7 days of culture, as shown in Figure 3.

The process of collecting and purifying cPGCs from the bloodstream involves several technically demanding steps. This poses a risk of even further loss of cells in the low population of available PGCs. Therefore, we stained the population of bloodstream cells with a fluorescent label immediately after their collection, prior to placing the cells in the culture condition. The characteristic cell behavioural differences allowed distinguishing the somatic blood cells, including erythrocytes, from cPGCs. Many of the seeded cPGCs were loosely connected to the fibroblast cell layer. The embryonic blood cells, however, remained suspended in the medium and did not adhere to the feeder layer. This resulted in blood cells being washed out after several medium changes. As shown in Figure 3, only the clusters of cPGCs remained on the feeder layer. This shows that the cultivation of whole blood cells labeled with PKH26 enabled to “selectively” grow cPGCs on the chicken feeder, without any prior purification step, but with the application of only specific culture conditions. We observed that the morphology was retained upon subculture and culture on the feeder layer. Small clusters were formed which is typical [28]. In the present study, the cultivated gPGCs and cPGCs migrated into the developing gonads of the recipient embryos (Figure 4 and Figure 6, Table 1 and Table 2). These results indicate that the pre-cultivated PGCs retained the ability to migrate to the recipient’s gonads as did the native PGCs. The doubling time for PGCs in this analysis was approximately 84 h. Woodcock et al. [2] reported that for cPGCs cultured in vitro, the doubling time was 33.4 h for both sexes. The cell doubling time depends on the genotype and is also affected by the concentration of the initiatory cell population. Different amounts of cells modify the culture conditions in vitro at various rates [28].

Conditions used in this research enabled us to cultivate both types of cells—cPGCs and gPGCs. Very few studies have compared the growth of both types of PGCs in the same culture condition. Recently, only Raucci et al. [11] compared three types of cells in a similar condition: Chicken embryonic stem cells (cES), cPGCs, and gonadal germ cells (GGCs). They used the same embryonic stem avian complete medium with differing FGF concentrations; moreover, an irradiated STO feeder layer was used only for cPGCs. Their conditions enabled a long-term culture of all stem cell types (over 30–40 days) and did not affect the reactivity to specific markers, but only a higher number of cytoplasmic bridges was observed in the gonadal population [29].

Chicken donor PGCs can be considered functional if, after being subjected to in vitro handling involving their cultivation, genetic modifications, and/or conservation protocols, they retain the contribution to germline. In other words, the following criteria should be met in functional PGCs: (1) Ability to colonize recipient gonads and differentiate into gametes, and (2) contribute to the germplasm of the produced chimeras and transmit the genes through generations. Therefore, it is a standard to include a validation task for at least one of the above functions. Here, we showed that both gPGCs and cPGCs that were pre-cultivated in a chicken feeder culture system retained their migratory properties and were incorporated into the recipient gonads. In this study, both gonads of recipient embryos (left and right) were colonized by gPGCs and cPGCs (Table 1 and Table 2). One may consider this system to be highly useful, because it allows cultivating both PGC types in the same condition. Naito [30] hypothesized that the population of cPGCs isolated from the embryonic bloodstream is heterogeneous and comprises stem cells with varying potential to differentiate. Therefore, it is expected to have increased differentiation of cPGCs in long-term cultures with loss of stem markers and reduced progenitor potential to form functional gametes. Here, some differences in the distribution of PGCs to the left and right gonads were observed using a subjective scale, e.g., for cPGCs, the right gonad might be considered a more preferred site (++ vs. +, Table 2). However, our results may elicit some contradicting arguments related to the established knowledge of the preferred site for gonadal colonization [22]. Thus, on the basis of our results, we cannot give a clear evidence of whether the sex of the recipient or the location of a gonad (left or right) influenced the fate of migration of the injected PGCs. This must be confirmed using a quantitative experimental design with a larger cohort of pre-cultivated donor PGCs.

## 5. Conclusions

In the present study, we proposed a method that enabled the cultivation of both cPGCs and gPGCs on the feeder layer of avian origin, with minimum xenobiotic factors. The migratory attraction to the recipient gonads was retained in both cPGCs and gPGCs that were pre-cultivated on the avian feeder layer. We also demonstrated a feasible and cost-efficient technique to routinely assess the cultivated PGCs on the basis of their morphological characteristics and using the optical features of PGCs in darkfield illumination. This method is especially useful to distinguish PGCs in the feeder culture systems or during co-cultivation of PGCs with other cell types. This can be supportive when the other popular protocols such as the MTT metabolic assay (a colorimetric assay for assessing cell metabolic activity) or an automated real-time cell counting system must be excluded due to issues related to the separation of heterogenic cell types in the PGC cultures.

Bearing in mind the mechanisms of epigenetic changes in PGCs [19,31] and factors affecting the molecular stability and fate of cultivated PGCs, further marker-assisted assays will be applied to our in vitro systems, followed by in vivo validation of the function of retained PGCs.

## 6. Patents

Patent application number: P.434242.

## Figures and Tables

**Figure 1 animals-10-01769-f001:**
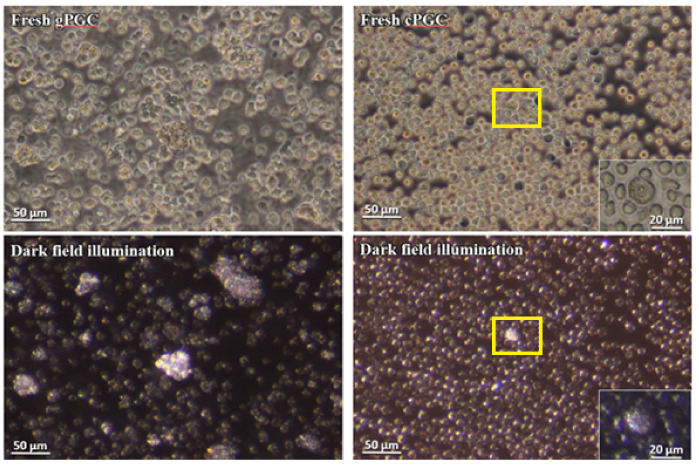
Fresh gonadal primordial germ cells (gPGCs) and circulating blood primordial germ cells (cPGCs) observed under an inverted phase-contrast microscope; gPGCs and cPGCs are visualized in darkfield illumination. The PGCs are distinguished as bright, illuminated cells among other cell types (P.434242 patent application number). Magnification with 40× objective (Leica DMI8, Germany).

**Figure 2 animals-10-01769-f002:**
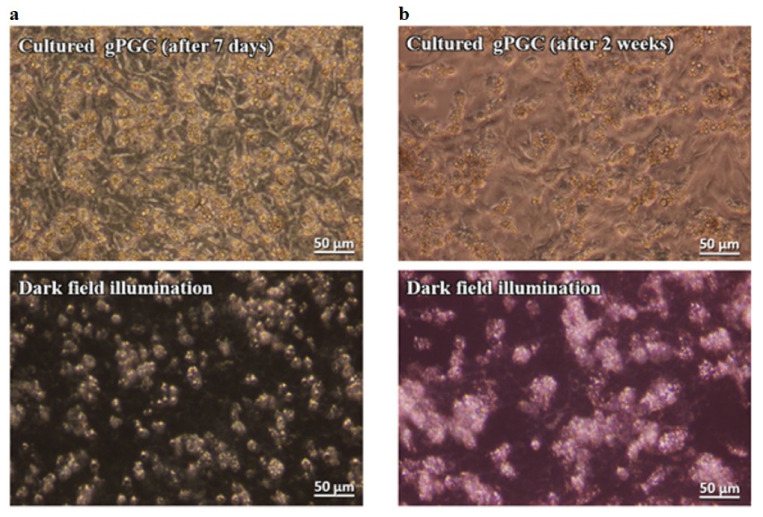
The gPGCs cultured for 7 and 14 days, observed under an inverted phase-contrast microscope and in darkfield illumination. This technique allows us to clearly distinguish PGCs from the feeder layer and other cells. The PGCs are visible as bright, illuminated cells among the other cell types (P.434242 patent application number). Magnification with 40× objective (Leica DMI8).

**Figure 3 animals-10-01769-f003:**
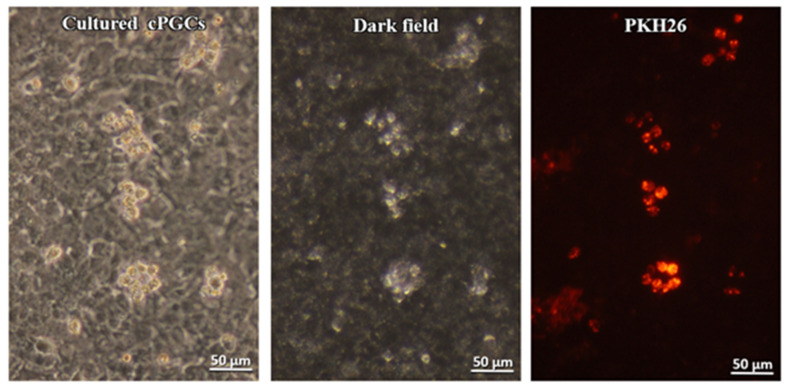
The cPGCs cultured for 7 days observed under an inverted phase-contrast microscope and darkfield illumination (P.434242 patent application number). The cells were stained after 7 days of culture using PKH26 fluorescent dye. Magnification of the objective 40× (Leica DMI8).

**Figure 4 animals-10-01769-f004:**
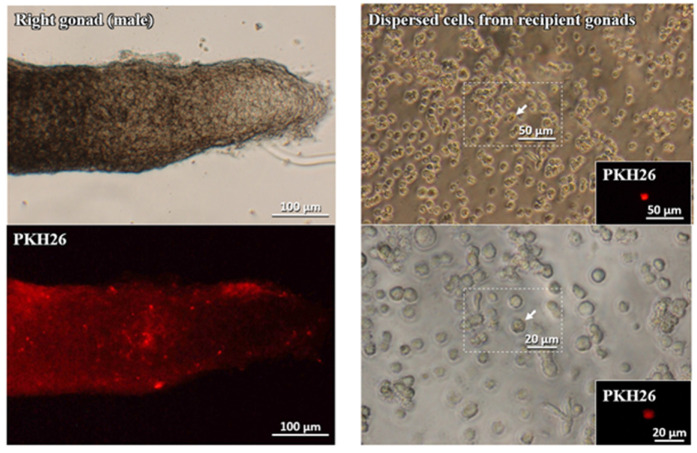
Left panel: Male gonad observed under the microscope (Leica DMI8) and stained with PKH26. Right panel: Dispersed cells from the recipient gonads and stained with PKH26.

**Figure 5 animals-10-01769-f005:**
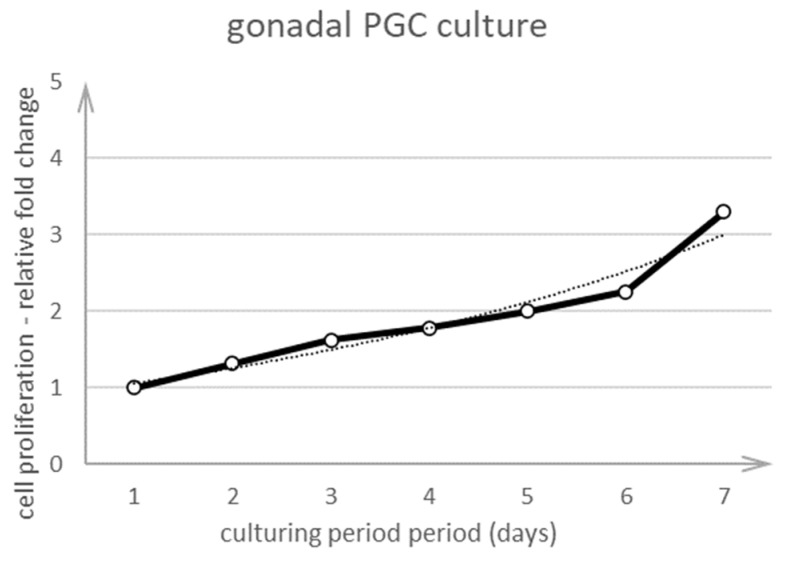
Proliferation curve of gPGCs cultivated on a chicken embryo feeder layer at pH 8.0. The live gPGCs cells were counted directly in the culture by using the darkfield illumination method.

**Figure 6 animals-10-01769-f006:**
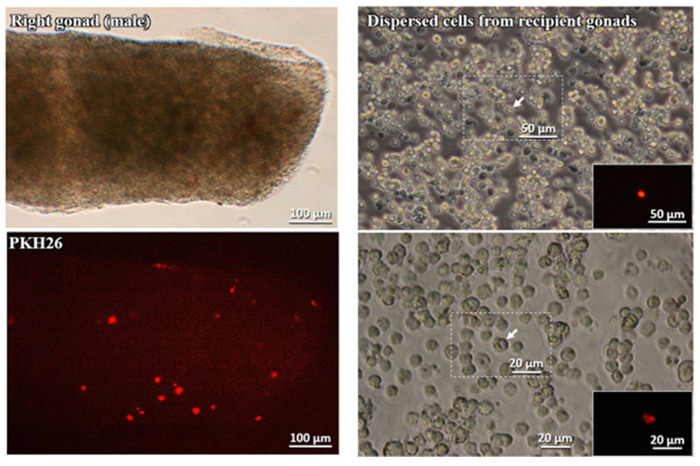
Left panel: The male gonad observed under the microscope (Leica DMI8) and stained with PKH26. Right panel: Dispersed cells from the recipient gonads and stained with PKH26.

**Figure 7 animals-10-01769-f007:**
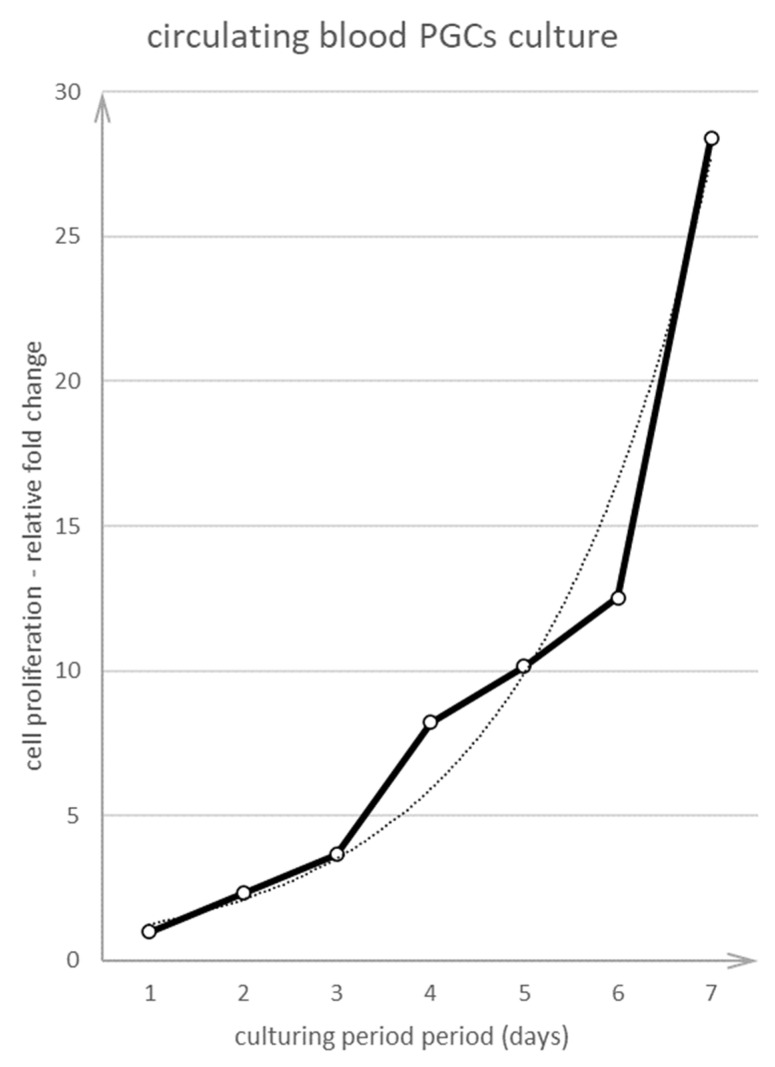
Proliferation curve of cPGCs cultivated on a chicken embryo feeder layer at pH 8.0. The live cPGCs were counted directly in the culture by using the darkfield illumination method.

**Table 1 animals-10-01769-t001:** Colonization of the recipient gonads with the cultivated gPGCs examined after 5 days post injection.

No. of Recipient Embryo (Injected Stages)	Sex	Presence of Injected gPGCs in Gonads	
		Right Gonad	Left Gonad
G-01 (stage 14)	Female	−	+
G-02 (stage 14)	Male	−	+
G-03 (stage 14)	Male	−	+
G-04 (stage 15)	Female	+	+
G-05 (stage15)	Female	−	+
G-06 (stage 15)	Male	+	+
G-07 (stage 16)	Female	++	+
G-08 (stage 15+)	Dead		
G-09 (stage 16)	Female	+	+
G-10 (stage 15)	Female	+	+
G-11 (stage 15)	Male	+	++
G-12 (stage 16)	Dead		
G-13 (stage 16)	Dead		
G-14 (stage 16)	Dead		

*Table legend*. G 01–14 = recipient embryos that received gPGCs; ‘−’ no colonization; ‘+’ gonad colonized at the moderate level; ‘++’ highly colonized gonad.

**Table 2 animals-10-01769-t002:** The recipient gonads with the cultivated cPGCs examined after 5 days post injection.

No. of Recipient Embryo (Injected Stages)	Sex	Presence of Injected gPGCs in Gonads	
		Right Gonad	Left Gonad
C-01 (stage 14)	Dead		
C-02 (stage 14)	Dead		
C-03 (stage 15)	Dead		
C-04 (stage 15)	Female	+	+
C-05 (stage 13)	Dead		
C-06 (stage 14)	Male	+	++
C-07 (stage 15)	Dead		
C-08 (stage 15)	Dead		
C-09 (stage 14)	Dead		
C-10 (stage 14)	Male	++	+
C-11 (stage 13+)	Female	++	+
C-12 (stage 13)	Male	−	−
C-13 (stage 16)	Male	−	−
C-14 (stage 15)	Female	++	−
C-15 (stage 14)	Male	+	+
C-16 (stage 16)	Male	++	+
C-17 (stage 16+)	Male	++	+
C-18 (stage 15+)	Dead		
C-19 (stage 15)	Dead		
C-20 (stage 16)	Female	+	+

*Table legend*. C 01–20 = recipient embryos that received cPGCs at the indicated stage 14–16+ H&H; ‘−’ no colonization; ‘+’ gonad colonized at the moderate level; ‘++’ highly colonized gonad.

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
