# Peer review of "In Vitro Culture of Chicken Circulating and Gonadal Primordial Germ Cells on a Somatic Feeder Layer of Avian Origin"

_animals, 2020, doi:10.3390/ani10101769_

Round 1

Reviewer 1 Report

The authors established in vitro culture system of chicken circulating and gonadal primordial germ cells on a feeder layer of chicken somatic cells. The method is very simple and well documented on the manuscript.

Minor points
L10 Primordial germ cells -> Chicken primordial germ cells (correct ?)
L25 There are two types of PGCs in chicken:
L45 to the next generations.
L50 low and different, for both types of PGCs contain lipid droplets and accumulated glycogen [6, 7].
L51 circulating cPGCs -> cPGCs (also L65, L85, L270, L277)
L51-52 (please add a reference)
L55 gonadal PGCs -> gPGCs (also L82)
L101 MMC -> mitomycin C, replace with L156
L222-228 should be included in Discussion section
L242 PGCs cells -> PGCs
L248 gPGCs clles -> gPGCs
L270 The circulating blood cPGCs cells -> The cPGCs
L272 cPGCs cells -> cPGCs
L277 circulating cPGCs -> cPGCs
L298 for his research. -> for their research.
L333 Woodcock et al. (2019) -> Woodcock et al. [2]
L355-356 please add a reference (Naito et al.)

Author Response

Thanks to the comments and suggestions of the Reviewer, the resulting manuscript is a great improvement over the text that was originally submitted and whatever the outcome.

Reviewer Comments to Author:

-Primordial germ cells -> Chicken primordial germ cells

Thanks to the comments and suggestions of the Reviewer, the resulting manuscript is a great improvement over the text that was originally submitted and whatever the outcome.

Reviewer Comments to Author:

  • L10 Primordial germ cells -> Chicken primordial germ cells

We corrected it.

  • L25 There are two types of PGCs in chicken:

We agree. We changed this fragment of the text.

  • L45 to the next generations.

We agree. We added to the text.

  • L50 low and different, for both types of PGCs contain lipid droplets and accumulated glycogen [6, 7].

We corrected it.

  • L51 circulating cPGCs -> cPGCs

We agree, we replaced “circulating cPGCs” with “cPGCs” in all text.

-  L51-52 (please add a reference)

We added a reference.

       - L55 gonadal PGCs -> gPGCs (also L82)

We  replaced “gonadal PGCs” with “gPGCs” in all text.

  • L101 MMC -> mitomycin C, replace with L156

As suggested, we replaced.

  • L242 PGCs cells -> PGCs

As suggested, we changed.

  • L248 gPGCs clles -> gPGCs

We corrected it.

  • L270 The circulating blood cPGCs cells -> The cPGCs

We  replaced.

  • L272 cPGCs cells -> cPGCs

We corrected it.

  • L277 circulating cPGCs -> cPGCs

We changed it.

L298 for his research. -> for their research.

Yes, we agree. We changed this fragment of the text.

  • L333 Woodcock et al. (2019) -> Woodcock et al. [2]

We changed it.

  • L355-356 please add a reference (Naito et al.)

We added a reference.

Reviewer 2 Report

In this study, authors describe the method to culture both cPGC and gPGC obtained from chicken embryos. Although the defined medium that support cPGC propagation in the absence of feeder cells has been reported (ref 16), this cannot support the proliferation of gPGC. Culture method by which both cPGC and gPGC can be grown might be useful to compare the characteristics of these cells. Furthermore, they propose the use of dark field illumination to simply discriminate PGCs from somatic cells. Simple and reliable method to confirm living PGC is useful in this field.

  1. Major concern is related to the characteristics of the PGCs. Since no purification step was involved, confirmation by typical staining method, such as PAS, SSEA-1 or CVH, is important.
  2. Please describe the contribution of the implanted PGC toward next generation, if you have any results.
  3. It may be useful if direct comparison of the differences in pH on the proliferative capacity is described.

Author Response

Thanks to the comments and suggestions of the Reviewer, the resulting manuscript is a great improvement over the text that was originally submitted and whatever the outcome.

Thanks to the comments and suggestions of the Reviewer, the resulting manuscript is a great improvement over the text that was originally submitted and whatever the outcome.

Reviewer Comments to Author:

  1. Major concern is related to the characteristics of the PGCs. Since no purification step was involved, confirmation by typical staining method, such as PAS, SSEA-1 or CVH, is important.

The PGCs were observed under an inverted phase-contrast microscope (Axiovert 40 VFL;

Zeiss, Germany). The cPGCs can be easily identified by their large size (12–16 μm in diameter), cytoplasm rich in yolk granules, and presence of spherical large-sized nuclei. Additionally, to  confirm  the  identity  of  PGCs,  we  also  performed the periodic  acid–Schiff  (PAS)  staining (date not presented). Finally, we used our simple and rapid method to identify live PGCs as bright cells under darkfield illumination. The PGCs rich in lipid droplets in their cytoplasm highly contrasted with the co-cultured feeder layer and other cell populations in the culture.

  1. Please describe the contribution of the implanted PGC toward next generation, if you have any results.

In this study the migratory ability of cultivated cells we tested on the level of recipient embryo only (Figures 4 and 6, Tables 1 and 2).

  1. It may be useful if direct comparison of the differences in pH on the proliferative capacity is described."

We did not conduct such research in this study.

Round 2

Reviewer 1 Report

The manuscript has been revised well. I think this manuscript will be acceptable now.

Reviewer 2 Report

I can confirm the reliability of the PGC characteristics by the authors' responses.